

# Vertical changes in volatile organic compounds (VOCs) and impacts on photochemical ozone formation

Xiao-Bing Li[1], Bin Yuan[1,*], Yibo Huangfu[1], Suxia Yang[2], Xin Song[1], Jipeng Qi[1], Xianjun He[1], Sihang Wang[1], Yubin Chen[1], Qing Yang[1], Yongxin Song[1], Yuwen Peng[1], Guiqian Tang[3,4], Jian Gao[5], Min Shao[1]

[1] College of Environment and Climate, Institute for Environmental and Climate Research, Guangdong-Hongkong-Macau Joint Laboratory of Collaborative Innovation for Environmental Quality, Jinan University, Guangzhou 511443, China

[2] Guangzhou Research Institute of Environment Protection Co., Ltd., Guangzhou 510620, China

[3] State Key Laboratory of Atmospheric Environment and Extreme Meteorology, Institute of Atmospheric Physics, Chinese Academy of Sciences, Beijing 100029, China

[4] University of Chinese Academy of Sciences, Beijing, 100049, China

[5] State Key Laboratory of Environmental Criteria and Risk Assessment, Chinese Research Academy of Environmental Sciences, Beijing 100012, China

[*] Corresponding author: Bin Yuan (byuan@jnu.edu.cn)





## Abstract

Volatile organic compounds (VOCs) play crucial roles in regulating the formation of tropospheric ozone. However, limited knowledge on the interactions between vertical VOC variations and photochemical ozone formation has hindered effective ozone control strategies, especially in large cities. In this study, we investigated the vertical changes in concentrations, compositions, and key driving factors of a large suite of VOCs using online gradient measurements taken from a 325 m tall tower in urban Beijing. We also analyzed the impact of these vertical VOC variations on photochemical ozone formation using box model simulations. Our results indicate that the vertical variations of various VOC species are strictly regulated by the diurnal evolution of the planetary boundary layer. During daytime, reactive hydrocarbons are rapidly oxidized as they mix upwards, leading to the formation of OVOCs. This process plays a more significant role in regulating photochemical ozone formation with increasing height. In the lower layer, the photochemical formation of ozone responds positively to changes in both NOx and VOCs. As a result, the production rate of ozone decreases with height due to significant reductions in the concentrations of both NOx and VOCs, but remains high in the middle and upper layers. The strong production of ozone aloft is primarily driven by high concentrations of OVOCs and hydroxyl radicals, which can act as an important source of ozone at ground level. Therefore, careful consideration should be given to the vertical variations in both photochemical ozone production rates and formation regimes in the whole boundary layer when developing regional ozone control strategies.



## 1 Introduction


Volatile organic compounds (VOCs) are crucial constituents of atmospheric
chemicals *(Li et al., 2022c)* and play important roles in regulating the atmospheric
oxidation capacity and contributing to the photochemical formation of tropospheric
ozone *(Zhao et al., 2022; Yang et al., 2024b)*. Ozone is a major air pollutant in urban
environments, with increasing trends reported globally over recent decades *(Fleming et*
*al., 2018; Cooper et al., 2020)*, despite stringent measures to control its precursor
emissions *(Wang et al., 2020b; Yeo and Kim, 2021; Li et al., 2022b; Perdigones et al.,*
*2022)*. As highlighted in previous studies, reducing emissions of reactive VOCs is key
to controlling ozone pollution at present and in the foreseeable future *(Zhao et al., 2022;*
*Wang et al., 2024)*.
The primary prerequisite for effective regional ozone pollution control is the
determination of the photochemical ozone formation regime *(Souri et al., 2020; Zhao*
*et al., 2022)*, which facilitates the development of reduction schemes for key precursor
emissions *(Ou et al., 2016; Wang et al., 2019)*. The main challenges in controlling
ozone pollution stem from the complex compositions of its precursors (e.g., VOCs and
NOx) in ambient air *(Guo et al., 2017; Wu et al., 2020; Li et al., 2022c)*, as well as the
complicated responses of photochemical ozone formation to changes in these
precursors *(Shao et al., 2021; Perdigones et al., 2022)*. Furthermore, the interactions
between vertical variations of ozone precursors and ozone formation remain unclear
*(Tang et al., 2017; Sun et al., 2018; Li et al., 2024)*, adding to the complexity of ozone
pollution control.
In most cases, the identification of key ozone precursors has been conducted using
ground-level observations *(Qi et al., 2021; Lu et al., 2022)* or compiled source emission
inventories *(Ou et al., 2015; An et al., 2021; Wang et al., 2022b)*. While these methods
are undoubtedly helpful in determining key ozone precursors and corresponding
reduction strategies, they often encounter unexpected uncertainties in urban regions.
*(Mo et al., 2018; Mo et al., 2020)*. Consequently, ground-level measurements of ozone





precursors have been favored to constrain model calculations *(Lu et al., 2012; Wang et*
*al., 2022a; Yang et al., 2022)* or provide empirical evidence for hypothesized theories
*(Hofzumahaus et al., 2009; Wang et al., 2022c)*. However, these ground-level
measurements cannot fully characterize atmospheric chemical processes in the entire
planetary boundary layer (PBL) due to strong vertical variations in precursor
concentrations *(Velasco et al., 2008; Li et al., 2018; Sun et al., 2018)*.
Ambient VOCs, as crucial ozone precursors, are composed of myriad species *(Wu*
*et al., 2020; Gkatzelis et al., 2021; Ye et al., 2021; He et al., 2022)* and serve diverse
functions in photochemical ozone formation *(Vo et al., 2018; Li et al., 2022a; Zhang et*
*al., 2022)*. Owing to the impact of variations in emission sources, chemical removal,
advection and convection transport, and secondary formation, the concentration and
composition of VOCs typically display notable vertical variability within the PBL,
especially in urban areas *(Li et al., 2022c)*. The ozone formation regime like undergoes
significant transitions from the ground to the upper boundary layer *(Li et al., 2024; Liu*
*et al., 2024a)*. Ozone generated throughout the PBL can influence surface ozone levels
due to enhanced atmospheric vertical mixing during the day. Consequently, it is
imperative to comprehend the vertical variations and principal determinants of VOCs,
as well as their effects on photochemical ozone formation within the PBL.
With the rapid development of cities in the recent two decades in China, a large
number of pollution-emitting industries and factories have been relocated from city
centers to alleviate air pollution. Concurrently, there's been a swift increase in the
ownership of electric vehicles *(Guo et al., 2021)*. These shifts in energy consumption
have driven the change in concentrations and compositions of VOCs in major cities like
Beijing *(Liu et al., 2024b)*, subsequently affecting photochemical ozone formation
*(Wang et al., 2024)*. However, the vertical variations and key drivers of VOCs and their
impacts on photochemical ozone formation in the urban PBL remain elusive. A primary
hurdle in studying these vertical changes in photochemical ozone formation is the
scarcity of reliable vertical VOC data *(Dieu Hien et al., 2019; Li et al., 2022c)*.
Engaging in vertical profiling of VOCs, ensuring all necessary species represented and



obtaining sufficient sample size, is especially challenging in the lower PBL where
atmospheric chemical reactions are most intense *(Benish et al., 2020; Kim et al., 2021)*.
Previous studies on vertical distributions of photochemical ozone formation in the
PBL have been conducted using measurements of a limited number of VOC species
and samples *(Zhang et al., 2018; Benish et al., 2020; Geng et al., 2020)*. In this study,
online gradient measurements of ozone, NOx, and a large suit of VOCs were made on
a 325 m tall tower in urban Beijing during the summer of 2021. Additionally, box model
simulations constrained by the gradient measurements were performed to analyze the
vertical variations and key drivers of VOCs as well as their impacts on photochemical
ozone formation.

## 2   Methods and materials

### 2.1   Description of the site, instrument, and field campaign

The data utilized in this study was derived from an intensive field campaign
conducted at the Beijing Meteorological Tower (BMT: 39°58′ N, 116°23′ E) between
July 6 and August 4, 2021. The BMT has a height of 325 m and is located in the northern
part of downtown Beijing, positioned between the third and fourth ring roads (Fig. S1).
A vertical observation system, established using long perfluoroalkoxy alkane (PFA)
Teflon tubes (OD: 1/2 in.), was used to make online gradient measurements of ozone,
NOx, and a set of VOCs on the BMT. Five specific heights, namely 15, 47, 102, 200,
and 320 m above ground level, were selected to mount the tube inlets, as depicted in
Fig. S2. An additional inlet, situated approximately 5 m above ground level, was
mounted on the rooftop of the observation room that was adjacent to the tower.
Consequently, the vertical observation system totally included a total of six sampling
inlets. The sampling inlet at the 15 m height was not utilized during this field campaign.
Filters were installed downstream of the tubing inlets on the tower to remove fine
particles. A rotary vane vacuum pump was used to simultaneously and continuously
draw sample air from the five tubes, ensuring that all tubes were flushed by ambient air



to reduce tubing delays of sticky organic compounds *(Pagonis et al., 2017; Liu et al.,*
*2019)*. Five critical orifices were employed to control the flow rate of the air stream in
each tubing, resulting in flow rates ranging between 15 and 20 standard liter per minute
(SLPM). Instruments drew sample air from the five tubes sequentially through a Teflon
solenoid valve group at designated time intervals. The switching time intervals of the
Teflon solenoid valve group were set as 4 minutes during this field campaign. The
measurements of trace gases in the first and last 1 minute of a four-minute period were
discarded to eliminate cross interferences between different inlet heights. Detailed
information on the vertical observation system and the assessment of trace gas
measurements through hundreds of meters long PFA tubes has been provided in our
previous works *(Li et al., 2023; Yang et al., 2024a)*.

Ozone was measured using the ultraviolet photometry method (49i, Thermo Fisher

Scientific Inc., USA). NO, $NO_2$, and NOx were measured using the chemiluminescence
method (42i, Thermo Fisher Scientific Inc., USA). Gradient measurements of ozone
and NOx were conducted at a time resolution of 10 seconds. The photolysis frequencies
of $NO_2$, represented by $j(NO_2)$, were measured by a spectrometer (PFS-100, Focused
Photonics Inc., China) situated on the rooftop of the observation room and have a time
resolution of 8 seconds. In situ measurements of meteorological parameters including
wind speed, air temperature, and relative humidity were made at 15 heights between 8
m and 320 m on the BMT with a time resolution of 20 seconds. Planetary boundary
layer height (PBLH) was obtained from the Air Resources Laboratory
(*https://ready.arl.noaa.gov/READYamet.php, last access: 10 June 2024*) and was
linearly interpolated to hourly values based on the initial time resolutions of three hours
*(Li and Fan, 2022)*.

A high-resolution proton-transfer-reaction quadrupole interface time-of-flight

mass spectrometer (PTR-ToF-MS, Ionicon Analytik, Austria) was employed to measure
VOCs at a time resolution of 10 seconds. The PTR-ToF-MS used both hydronium ion
($H_3O^+$) *(Yuan et al., 2017; Wu et al., 2020; Li et al., 2022c)* and nitric oxide ion ($NO^+$)
*(Wang et al., 2020a)* as reagent ions. These two reagent ions were automatically



switched every 60 min for $H_3O^+$ and every 22 min for $NO^+$ throughout the campaign.
The PTR-ToF-MS operated at an E/N value of approximately 120 Td in $H_3O^+$ mode
and an E/N value of around 60 Td in $NO^+$ mode. Instrument backgrounds were
automatically measured during the last two minutes of each operation mode by passing
ambient air through a platinum catalyst heated to 365 ℃. A gas standard containing 39
VOC species was used to calibrate the PTR-ToF-MS daily. Sensitivities for the
remaining species were determined based on reaction kinetics of the PTR-ToF-MS *(Wu*
*et al., 2020)*. Impacts of ambient humidity on the PTR-ToF-MS measurements were
corrected by using humidity-dependence curves of VOCs obtained in our laboratory
*(Wang et al., 2020a; Wu et al., 2020)*. Carbon dioxide ($CO_2$ in dry air) and humidity
were measured using a $CO_2$ and $H_2O$ gas analyzer (Li-840A, Licor Inc., USA) at a time
resolution of 10 seconds.

Gradient measurements of the total OH reactivity (OHR) of atmospheric trace

gases were made using the improved comparative reactivity method (ICRM) developed
by our team *(Wang et al., 2021a)* from July 28 to 31. In addition, gradient measurements
of carbon monoxide (CO), methane ($CH_4$), $CO_2$, and $H_2O$ were simultaneously
measured using the cavity ring-down spectroscopy (CRDS) method (G-2401, Picarro
Inc., USA) at a time resolution of 10 seconds from May 15 to June 25. Sulfur dioxide
($SO_2$) was measured using the ultraviolet fluorescence method (43i, Thermo Fisher
Scientific Inc., USA) at a time resolution of 10 seconds from June 25 to August 3. The
total OHR of VOCs, denoted by $OHR_{VOCs}$, can be estimated by excluding those of the
inorganic species (namely ozone, NOx, CO, $SO_2$, and $CH_4$). It should be noted that
gradient measurements of $CH_4$ and CO were not made during July 28-31, and their
average concentrations in daytime (11:00-16:00 LT) between May 15 and June 25 at 5
m were used for all altitudes to calculate $OHR_{VOCs}$. This method will bring minor
uncertainties due to the minor vertical differences in concentrations of $CH_4$ and CO in
daytime (Fig. S3). The OHR of VOCs can also be calculated by summing the products
of their measured concentrations and their reaction rate coefficients with OH radicals,
as formulated in Eq. (1):



$$OHR = \sum k^i_{OH-R}[VOC_i] \qquad \text{Eq. (1)}$$

where $k^i_{OH-R}$ is the reaction rate coefficient of the $i^{th}$ VOC species with OH radical
and $[VOC_i]$ is the concentration of the $i^{th}$ VOC species.

## 2.2 Estimation of NMHC concentrations at the BMT site

The PTR-ToF-MS is limited in its ability to measure VOC species with proton
affinities higher than $H_2O$ (691 kJ mol$^{-1}$) when operating in the $H_3O^+$ mode *(Yuan et*
*al., 2017)*. This limitation results in the absence of certain nonmethane hydrocarbons
(NMHCs), such as alkanes and many alkene species, which play important roles in
photochemical ozone formation. To obtain a comprehensive understanding of vertical
variations in concentrations, compositions, and environmental impacts of VOCs, this
study estimated the vertical profiles of those unmeasured NMHC species based on the
concentrations of measured VOCs using the PTR-ToF-MS. Detailed information on
estimation of NMHC concentrations is provided in SI.

## 2.3 Box model setup

A zero-dimension box model (F0AM) coupled with the Master Chemical
Mechanism (v3.3.1) *(Wolfe et al., 2016; Yang et al., 2022)* was used to compute the
production rate of ozone, denoted by P(O$_3$) as formulated in Eq. (2):

$$P(O_3) = k_{HO_2+NO}[HO_2][NO] + \sum k^i_{RO_2+NO}[R^iO_2][NO] \qquad \text{Eq. (2)}$$

$$k_{RO_2+NO}[RO_2][NO]$$
where [HO$_2$] and [NO] is the concentrations of HO$_2$ and NO, [R$^i$O$_2$] is the concentration
of the $i^{th}$ organic peroxyl radical. The relative incremental reactivity (RIR) of
photochemical ozone production to changes in different precursors was determined
using Eq. (3):

$$RIR(X) = \frac{\left[P^S_{O_x}(X) - P^S_{O_x}(X - \Delta X)\right]/P^S_{O_x}(X)}{\Delta S(X)/S(X)} \qquad \text{Eq. (3)}$$



where X represents ozone precursors, $P_{O_x}^S(X)$ is the contribution of X to the production
rate of Ox, $\Delta X$ is the amount of change in ozone precursors, S(X) is the initial
concentration of X. RIR values were used to discern sensitivities of photochemical
ozone formation to changes in precursor gases. A positive RIR(X) value suggests that
an increase in X enhances ozone formation, while a negative RIR value indicates that
an increase in X inhibits ozone formation.

Model calculations were constrained by measurements of ozone, NOx, CO, a suit

of VOCs, air temperature, and relative humidity. In addition to the measured or
estimated concentrations of NMHCs, nine oxygenated VOC (OVOC) species (Table
S1) measured by PTR-ToF-MS were used to constrain the model calculation. The
model was run in a time-dependent mode with a time resolution of 5 minutes and a
spin-up period of 2 days *(Lu et al., 2012; Wang et al., 2022c)*. The dry deposition
velocity of ozone was set as 0.27 cm s$^{-1}$ when calculating P(O$_3$) 5 m and was zeroed
out when calculating P(O$_3$) at other heights.

## 3   Results and discussions

### 3.1   Temporal and vertical variations in concentrations of trace gases

As shown in Fig. 1, the meteorology in Beijing was characterized by high air

temperature (27.3±2.9 °C), high humidity (83.9%±16.2%), and gentle winds (1.1±0.4
m s$^{-1}$) throughout the campaign. The intense solar radiation, elevated air temperature,
and mild winds favored the photochemical formation and accumulation of ozone,
leading to frequent occurrences of ozone pollution episodes. Fig. 1 also presents time
series of mixing ratios of ozone and its selected precursors (namely isoprene, toluene,
monoterpenes, and NOx) along with *j*(NO$_2$) measured at 5 m. The campaign mean
ozone mixing ratio was 45.6±25.3 ppb, but the maximum hourly mean ozone mixing
ratio reached 129.3 ppb, indicating strong photochemical reactions in urban Beijing
during the campaign. Surface ozone concentrations exhibited a typical diurnal variation



pattern with the maximum occurring at 16:00 LT (Fig. S5), implying its predominant
source from local photochemical production.
Isoprene is a typical tracer of biogenic emissions and is also a highly reactive VOC
species *(Atkinson and Arey, 2003)*. Isoprene had a campaign mean mixing ratio of
$0.7\pm0.6$ ppb. The average diurnal profile of isoprene at 5 m has a unimodal pattern with
the maximum occurring at 14:00 LT (Fig. S5), exhibiting strong dependence on solar
radiation. Monoterpenes were also generally recognized as typical tracers of biogenic
emissions *(Gómez et al., 2020)* and have a campaign mean mixing ratio of $0.3\pm0.3$ ppb.
The average diurnal profile of monoterpenes was characterized by low mixing ratios in
daytime with two peaks occurring at 05:00 and 20:00 LT, respectively.
Toluene and NOx are recognized as typical tracers of anthropogenic emissions in
urban regions *(Niu et al., 2017; Li et al., 2022c)*, with campaign mean mixing ratios of
$0.7\pm0.7$ and $8.1\pm4.8$ ppb, respectively. The average diurnal profiles of toluene and NOx
at 5 m exhibited similar variations with larger values at night than during the day. Based
on the measured concentrations and diurnal variations of ozone and its key precursors
at ground level, it can be inferred that urban Beijing is experiencing severe ozone
pollution, which is predominantly contributed by local photochemical production. As
key ozone precursors, ambient concentrations of VOCs are contributed by the mixture
of anthropogenic and biogenic sources.
Fig. 2 shows the average diurnal and vertical variations in mixing ratios of ozone,
NOx, Ox ($O_3+NO_2$), and six selected VOC species (three hydrocarbons and three
OVOCs) within the measurement height range of 5-320 m. High mixing ratios of ozone
were observed in the afternoon following the enhancement of solar radiation, which
was consistent with the diurnal change pattern of ozone concentrations at the ground
level. The vertical gradients of ozone mixing ratios were positive throughout the day
but substantially enhanced at night (Fig. 3). The lower ozone mixing ratios near the
surface than aloft were mainly caused by the enhancement of dry deposition and NO
titration *(Brown et al., 2007; Ma et al., 2013; Li et al., 2022b)*.





NOx is a primary pollutant and mainly contributed by vehicular exhausts in urban
regions. In contrast to ozone, NOx mixing ratios were low in daytime and exhibited
negative vertical gradients throughout the day, as shown in Figs. 2B and 3A-B. In
nighttime, large amounts of local NOx emissions were trapped and accumulated in a
shallow boundary layer (<100 m). NOx concentrations rapidly decreased with height
even in the overlying residual layer due to the suppression of turbulence vertical mixing.
With the onset of sunlight, the PBL rapidly expanded due to the surface heating effect.
The accumulated high concentrations of NOx in the shallow nocturnal boundary layer
were thereupon diluted and removed by photochemical reactions.
Ox is frequently used as a conserved metric to investigate temporal and spatial
variability of ozone by eliminating the NO titration effect. As shown in Fig. 2C, the
mixing ratios of Ox had similar diurnal and vertical variation patterns to those of ozone,
but the vertical gradients of Ox were weaker than those of ozone. This result suggests
that the vertical distribution of NO concentrations played an important role in regulating
the vertical change of ozone concentrations. The enhanced positive gradients of ozone
mixing ratios at night were predominantly due to the strict suppression of turbulence
vertical mixing *(Geyer and Stutz, 2004)*. The higher concentrations of ozone aloft are
considered as the residual of the ozone produced in the daytime PBL and have been
recognized as an important reservoir for the enhancement of surface ozone in morning
periods *(Kaser et al., 2017; Li and Fan, 2022; He et al., 2023)*.
Benzene and toluene demonstrated similar diurnal and vertical variations to NOx,
with low concentrations in daytime and high concentrations at night, as shown in Figs.
2D-F and 3A-B. The concentrations of both benzene and toluene decreased with height
throughout the day, confirming their primary emissions from ground-level sources.
However, unlike benzene, the diurnal and vertical variations of toluene were more
pronounced. Isoprene emissions are highly dependent on solar radiation, resulting in its
higher concentrations in the early afternoon compared to other times of the day.
Isoprene mixing ratios also exhibited strong negative vertical gradients below 320 m
throughout the day. In contrast to toluene, isoprene concentrations decreased more





rapidly with height in the daytime. For instance, the mixing ratios of isoprene decreased
by approximately 70% from 5 to 320 m in the daytime, while it was only 30% for
toluene.

Fig. 3A-B show the average vertical profiles of the NMHCs, normalized to their

respective ground-level concentrations measured by the PTR-ToF-MS in daytime and
nighttime. The normalized mixing ratios of the NMHCs exhibited significantly
differentiated gradients in daytime. In contrast, apart from monoterpenes, the
differences in vertical gradients of the normalized vertical profiles for other NMHCs
were relatively small at night. The differentiated vertical gradients of the NMHCs in
daytime were primarily caused by their intrinsic chemical reactivities, such as reactions
with OH radicals. As shown in Fig. 4, concentration ratios of the NMHC species
between 320 m and 5 m with $k_{OH}$ values lower than $2.5 \times 10^{-11}$ cm$^{-3}$ molecule$^{-1}$ s$^{-1}$
exhibited slight variability and rapidly declined with the further increases in $k_{OH}$. The
lower NMHC concentrations at higher altitudes were predominantly caused by the
combined effects of atmospheric diffusion and chemical removal *(Sangiorgi et al.,*
*2011).*

Considering the effects of atmospheric diffusion and chemical removal by

reactions with OH radicals, concentration ratios of NMHC species between 320 m and
5 m in daytime can be estimated using Eq. (4):

$$y = A \times \exp(-k_{OH}[OH]\Delta t) \qquad\qquad \text{Eq. (4)}$$

where *y* represents concentration ratios of the NMHC species between two altitudes, A
represents the effect of atmospheric dilution, $k_{OH}$ is the reaction rate coefficient of
NMHCs with OH radicals, [OH] is the concentration of OH radical, $\Delta t$ is the
turbulence mixing time scale between the two altitudes. The term $[OH]\Delta t$ thus refers
to the exposure of NHMCs to OH radicals between the two altitudes. As shown in Fig.
4, the average concentration ratios of NMHCs between 320 m and 5 m in daytime
during the campaign can be well reproduced using Eq. (4) with the coefficients A of
0.88 and $[OH]\Delta t$ of $1.0 \times 10^{10}$ molecules cm$^{-3}$ s. Atmospheric diffusion processes have
same impact on the vertical distributions of all trace gases. The differences in vertical





gradients of NMHCs were mainly determined by the differences in their chemical
removal rates without considering influences from advection transport.

Methanol, as one of the most abundant OVOC species in the atmosphere, had its

lowest concentrations during daytime and displayed negative vertical gradients
throughout the day, as shown in Fig. 2G. The vertical and diurnal variations of methanol
suggest that its ambient concentrations in urban Beijing were mainly contributed by
local primary emissions. Conversely, formaldehyde and MVK+MACR (the first-
generation oxidation products of isoprene), as the photochemical oxidation products of
NMHCs, had higher concentrations during daytime than at night and exhibited
relatively weak vertical concentration gradients (Fig. 2H-I). This is mainly because
these OVOCs are produced from the oxidation of NMHCs during turbulence vertical
mixing and will accumulate in high altitudes. These phenomena were also observed for
other OVOC species, as shown in Fig. 3.

The vertical and diurnal variations in concentrations of ozone, NOx, and VOCs

are intricately governed by their sources, chemical reactivities, and the evolution of the
PBL (namely the vertical dilution conditions). A significant accumulation of VOCs in
the shallow nocturnal PBL is subsequently vertically diluted and chemically removed
during daytime, thereby impacting the photochemical formation of ozone within the
daytime PBL. In addition, the observed vertical changes in concentrations of VOCs
imply that they will play distinct roles in contributing to photochemical ozone
formation.
**3.2   Vertical variations in contributions of VOCs to OHR**

During the daytime, VOCs are primarily oxidized by OH radicals and contribute

to the photochemical formation of ozone. To provide an overview on the vertical
variations in contributions of different VOCs to OHR, another 1204 ions measured by
the PTR-ToF-MS and can be quantified were used for analysis. All the VOCs were
classified into three large categories, namely $C_xH_y$ (including alkanes, alkenes,
aromatics, and other hydrocarbons; 121 species), OVOCs ($C_xH_yO_1$, 121 species;





$C_xH_yO_2$, 120 species; $C_xH_yO_{\geq3}$, 256 species), and N/S-containing (653 species), as
shown in Fig.5. Acetylene is included in alkenes.
Fig. 5A illustrates that the total mixing ratios of VOCs in daytime exhibited a slight
downward trend from 5 m to 320 m, primarily due to the rapid decrease in mixing ratios
of the $C_xH_y$ category. The total mixing ratios of the $C_xH_y$ category decreased from 16.8
to 10.6 ppb from 5 m to 320 m, with alkanes making the largest contribution, followed
by alkenes, aromatics, and other $C_xH_y$. Alkanes constituted 58% of the total mixing
ratios of $C_xH_y$ at 5 m, but this proportion increased to 65% at 320 m. The fractional
contributions of alkenes and aromatics in the total mixing ratios of $C_xH_y$ slightly
declined from 28% to 22% and from 12% to 10%, respectively, between these two
altitudes. As for OVOCs, the $C_xH_yO_1$ category was the most abundant among the
measurements, contributing to 52%-58% of the total mixing ratios at the five heights,
followed by the $C_xH_yO_2$ (8%-10%), and $C_xH_yO_{\geq3}$ (2%) categories. The mixing ratios of
the N/S-containing category slightly varied around 2.8 ppb between 5-320 m,
contributed to approximately 6% of the total VOC concentrations.
Similar to the vertical variations in concentrations, OHRs of the $C_xH_y$ category,
denoted by $OHR_{CH}$, also rapidly decreased from 6.9 s$^{-1}$ to 2.5 s$^{-1}$ between 5 and 320 m,
accounting for 52%-31% in the total OHRs of VOCs (Fig. 5B). Fractional contributions
of alkenes (40-18%), alkanes (5%), and aromatics (5%-4%) to the total OHRs of VOCs
all exhibited decreasing tendencies from 5 m to 320 m. The total OHRs of alkenes
decreased more quickly from 5 to 320 m than those of alkanes and aromatics. OHRs of
the other $C_xH_y$ category stabilized at approximately 0.3 s$^{-1}$ below 320 m, exhibiting an
increasing contribution (2%-4%) to the total OHRs of VOCs with the increase in height.
The OHRs of other VOC categories only slightly varied without exhibiting a clear
variation trend from 5 to 320 m during the day. As a result, fractional contributions of
the $C_xH_yO_1$ (27%-42%), $C_xH_yO_2$ (12%-18%), and $C_xH_yO_{\geq3}$ (5%-7%), and N/S-
containing (2%-4%) categories in the total OHRs of VOCs all increased with height.
The increased contributions of OVOCs and N/S-containing species to the total



concentrations and OHRs of VOCs implied that air masses became more aged with the
increase in height.

As depicted in Fig. 6A-B, high $OHR_{CH}$ values were mainly constrained in the PBL

and is mainly contributed by biogenic hydrocarbons, specifically isoprene, during
daytime due to their high OH reactivities and enhanced emissions. The fractional
contributions of isoprene in $OHR_{CH}$ decreased rapidly with increasing height (Fig. 7A).
For instance, isoprene accounted for a campaign median fraction of 58% in $OHR_{CH}$ at
5 m in daytime, making it a frequent contributor to photochemical ozone formation in
urban regions. However, this faction decreased to 38% at 320 m. Therefore, it can be
speculated that the total contributions of hydrocarbons to the total OHRs of VOCs will
also rapidly decline from 320 m to the top of the PBL, which typically ranges between
several hundreds of meters to approximately 2~3 km in daytime (Fig. S6).

The total concentrations and OHRs of OVOCs only slightly decreased with the

increase in height below 320 m in daytime, as shown in Fig. 6C-D. This is consistent
with the results of *(Wang et al., 2021b)*, which observed high concentrations of OVOCs
in the upper PBL. Consequently, the ratio of $OHR_{OVOC}$ to $OHR_{CH}$, denoted by
$OHR_{OVOC}/OHR_{CH}$, rapidly increased from 0.87 at 5 m to 2.6 at 320 m (Fig. 7A). This
suggests that OVOCs may play more important roles in regulating the photochemical
ozone formation in the middle and upper layers. To assess their potential roles in
contributing to the photochemical ozone formation throughout the PBL, we calculated
the mean OHRs (MOHR) of different VOC categories in daytime using Eq. (5):

$$MOHR(X) = (\sum([X]_i + [X]_{i-1})(h_i - h_{i-1})/2)/(320 - 5) \qquad \text{Eq. (5)}$$

where $MOHR(X)$ is the MOHR of the VOC category X, $[X]_i$ is the concentration of
X at the $i$th altitude (namely 5, 47, 102, 200, and 320 m for $h_i$) above ground level.

As shown in Fig. 7B, the campaign median MOHR for isoprene was 1.7 s$^{-1}$ and

accounted for 48% of the campaign median MOHR of the $C_xH_y$ category. This fraction
was significantly lower than that of isoprene (57%) in $OHR_{CH}$ at 5 m. In addition, the
campaign median MOHR of the $C_xH_y$ category (3.5 s$^{-1}$) was also significantly lower





than the $OHR_{CH}$ (6.0 s$^{-1}$) at 5 m. By contrast, the campaign median MOHR of OVOCs
(4.8 s$^{-1}$) was comparable to that of $OHR_{OVOC}$ (4.9 s$^{-1}$) at 5 m. As unsaturated
hydrocarbons, most alkene species are more reactive than alkanes and aromatics
*(Atkinson and Arey, 2003)*. As a result, alkenes had dominant contributions to the
MOHR of the $C_xH_y$ category and the $OHR_{CH}$ at 5 m in daytime. As shown in Fig. 7C,
the campaign mean OHRs of alkanes, alkenes, and aromatics at 5 m in daytime were
0.7, 5.2, and 0.7 s$^{-1}$, respectively, accounting for 10%, 75%, and 10% of the $OHR_{CH}$.
However, the campaign mean MOHRs of alkanes, alkenes, and aromatics were 0.5, 2.7,
and 0.5 s$^{-1}$, respectively, accounting for 12%, 68%, and 12% of the MOHR of NMHC.
We can also expect that the total contributions of alkenes to the MOHR of the $C_xH_y$
category in daytime will significantly decrease if their vertical distributions in the whole
PBL are considered.
This study investigated and compared the vertical profiles of measured $OHR_{VOCs}$
and calculated $OHR_{CH}$ during daytime over the period of July 28-31, as shown in Fig.
7D. The campaign median of the measured $OHR_{VOCs}$ exhibited a slow decrease from
38.4 s$^{-1}$ at 5 m to 25.4 s$^{-1}$ at 320 m. As anticipated, the $OHR_{CH}/OHR_{VOCs}$ ratio declined
rapidly from 16% to 7% from 5 to 320 m. It is important to note that the small
$OHR_{CH}/OHR_{VOCs}$ ratio and its declining trend with the increasing height do not imply
the insignificant roles of hydrocarbons in regulating the secondary pollutant formation
in higher altitudes. The measured concentrations of hydrocarbons are merely the
remnants of chemical reactions. The oxidation products of NMHCs, such as OVOCs
and organic nitrates, formed during vertical mixing in daytime, will continue to
participate in atmospheric chemical reactions.
**3.3  Vertical variations in photochemical ozone formation**
The surface ozone budget is intimately linked to the vertical variations of
photochemical ozone formation throughout the PBL. Previous studies have consistently
reported that the photochemical formation of ozone, encompassing both P(O$_3$) and
ozone formation regimes (namely the NOx-limited, VOCs-limited, and transition



regimes), are highly dependent on the change in its precursors *(Shao et al., 2021; Yang*
*et al., 2022)*. Consequently, any changes in the concentrations and compositions of
VOCs and NOx within the PBL will inevitably lead to alternations in the vertical
distribution of $P(O_3)$ and ozone formation regimes *(Tang et al., 2017; Li et al., 2024)*.
Fig. 8A illustrates the average dependence of $P(O_3)$ on NOx concentrations along
with the normalized probability density (NPD) distribution of NOx concentrations at 5
m, 200 m, and 320 m in daytime during the field campaign. At different heights, $P(O_3)$
all rapidly increased with the rise in NOx until a critical NOx mixing ratio was reached,
after which $P(O_3)$ decreased slowly. The critical NOx mixing ratios decreased from
approximately 9.5 ppb at 5 m to 5.0 m ppb at 320 m, primarily caused by the decreases
in both NOx concentrations and the OHRs of VOCs. As also shown in Fig. 8A, the
majority of the measured NOx mixing ratios fall into the transition zone of the $P(O_3)$-
NOx curves, suggesting that the photochemical ozone formation in Beijing belonged to
the transition regime below 320 m.
RIR values were also calculated using the box model results to further elucidate
the sensitivities of photochemical ozone formation to changes in multiple precursors at
different altitudes. As shown in Fig. 8B, positive RIR values were observed for both
NOx and various VOC groups at the five heights, further confirming that the
photochemical ozone formation belonged to the transition regime in the lower layer.
RIR values for NOx rapidly declined from 5 to 320 m, implying that the photochemical
ozone formation in higher altitudes were more prone to be controlled by the abundance
of VOCs. This is also manifested by the increasing RIR values for both AVOCs and
OVOCs from 5 m to 320 m. RIR values for BVOCs significantly decreased with height
due to their rapid removal by reactions with OH radicals when being vertically mixed.
These results are consistent with the results in section 3.3 that the less reactive AVOCs
and OVOCs are the dominant species in regulating the photochemical formation of
ozone in urban regions aloft.
According to the vertical distribution patterns of the photochemical ozone
formation regime, $P(O_3)$ decreases with increasing height alongside simultaneous



declines in concentrations of both NOx and VOCs. Fig. 8C presents the average diurnal
and vertical variations in $P(O_3)$ calculated by the box model during the campaign. The
$P(O_3)$ values were higher in daytime and correlated well with $j(NO_2)$. As shown in Fig.
S7, OH radical concentrations and $P(O_3)$ exhibited contrasting vertical distribution
patterns in daytime. $P(O_3)$ decreased from the ground to 320 m, where it still maintained
a relatively high value of approximately 10 ppb h$^{-1}$ at noon. These results highlight that
the photochemical formation of ozone aloft also remained strong compared to those at
ground level. Consequently, the downward transport of ozone from high altitudes,
driven by turbulence mixing, can become significant sources of surface ozone during
the day *(Karl et al., 2023)*.

Due to the measurement height limitation, the vertical distributions of $P(O_3)$ in the

middle and upper parts of the PBL were not determined in this study. As reported by
the work in *(Benish et al., 2020)*, $P(O_3)$ typically exhibited weak and nearly linear
decline tendencies from 300 m to the top of the PBL during daytime. $P(O_3)$ at the PBL
top was approximately half of that at 300 m. Consequently, we can assume that $P(O_3)$
decreased linearly from 320 m to the top of the PBL. The integral of $P(O_3)$ at different
heights within the PBL can then be estimated using a similar method as described in
Eq. (5).

As shown in Fig. 8D, the total amount of ozone photochemically produced below

47 m constituted a mere 6% of the entire PBL. This fractional contribution increased to
approximately 35% at 320 m, further corroborating that the majority of the boundary-
layer ozone was produced in the middle and upper layers. Given the enhancement of
turbulence vertical mixing in daytime, ozone produced at high altitudes becomes a
significant source of surface ozone. This is substantiated by the widespread reports of
strong downward ozone fluxes in the bottom part of the PBL (tens of meters above
ground level) *(Fares et al., 2010; Liu et al., 2021; Karl et al., 2023)*. Consequently,
when devising ozone control strategies, particularly in urban regions with intricate
precursor emissions, careful considerations should be given to the vertical variations in
the formation regimes of ozone in the PBL.



## 4 Conclusions

In this study, we investigated the vertical variations, key drivers, and environmental impacts of VOCs in the PBL using tower-based online gradient measurements in urban Beijing during the summer of 2021. The diurnal and vertical variations of various VOC species were strictly regulated by the diurnal evolution of the PBL. In daytime, reactive NMHC species were rapidly oxidized when they were mixed upward along with the formation of OVOCs. As a result, OVOC species played more significant roles in regulating the photochemical ozone formation in urban regions aloft. The photochemical formation of ozone belongs to the transition regime in the lower part of the PBL and becomes more sensitive to changes in the concentrations of AVOCs and OVOCs with increasing height. $P(O_3)$ exhibited decreasing tendencies with height but remained very large in high altitudes, likely driven by the high concentrations of OVOCs and OH radicals. Therefore, careful consideration should be given to the vertical variations in both $P(O_3)$ and photochemical ozone formation regimes in the whole PBL when making regional ozone control strategies.

The vertical variations in concentrations and compositions of VOCs significantly influence the formation of secondary pollutants. Furthermore, vertical changes in chemical reaction environments (e.g., temperature, humidity, and solar radiation) and concentrations of other chemicals (e.g., particulate matters, NOx, ozone) can also impact the degradation pathways of VOCs. These factors also affect the formation pathways and production yields of secondary pollutants. This is particularly crucial for the highly reactive NMHCs in urban areas with complex anthropogenic emissions and is expected to be thoroughly elucidated in future studies.

## Data availability

The observational data used in this study are available from corresponding authors upon request.



## Author contributions

BY, XBL, and YH designed the research. XBL, BY, YH, XS, JQ, XH, SW, YC, QY, YS, YP, GT, JG, and MS contributed to the data collection and data analysis. XBL, SY, and BY designed and performed the box model simulations. XBL and BY wrote the paper with contributions from all coauthors. All the coauthors discussed the results and reviewed the paper.

## Competing interests

The authors declare that they have no conflict of interest.

## Acknowledgments

The authors would like to thank the personnel who participated in data collection, instrument maintenance, and logistic support during the field campaign.

## Financial support

This work was financially supported by the National Key R&D Plan of China (grant nos. 2023YFC3706103, 2023YFC3706201, 2023YFC3710900, and 2022YFC3700604) and the National Natural Science Foundation of China (grant nos. 42121004, 42275103, 42205094, 42230701, and 42305095, 42475107). This work was also supported by the Special Fund Project for Science and Technology Innovation Strategy of Guangdong Province (grant no. 2019B121205004), Guangdong Basic and Applied Basic Research Foundation (grant no. 2024A1515011570) and Guangzhou Basic and Applied Basic Research Foundation (grant no. 2024A04J3958).

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

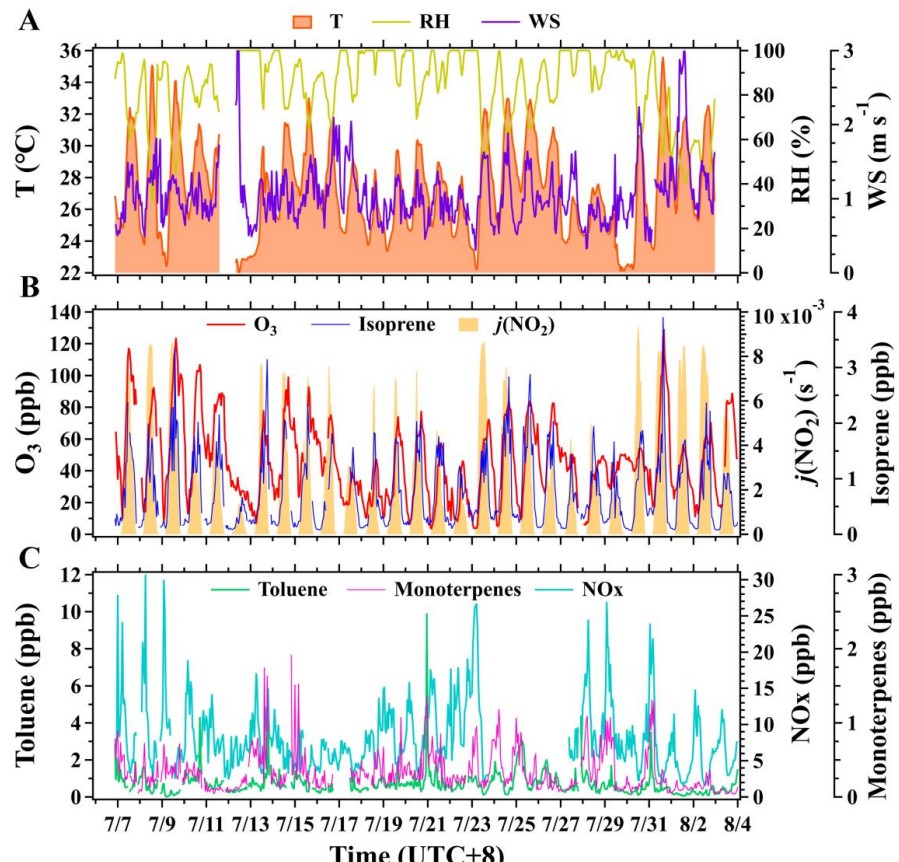


**Figure 1.** Time series of hourly mean air temperature (T), relative humidity (RH), wind

speed (WS), and mixing ratios of surface ozone, NOx, and VOC species along with

$j(NO_2)$ at the BMT site during the campaign. Meteorological parameters were measured

at 8 m above ground level and mixing ratios of ozone and its selected precursors were

measured at 5 m above ground level.

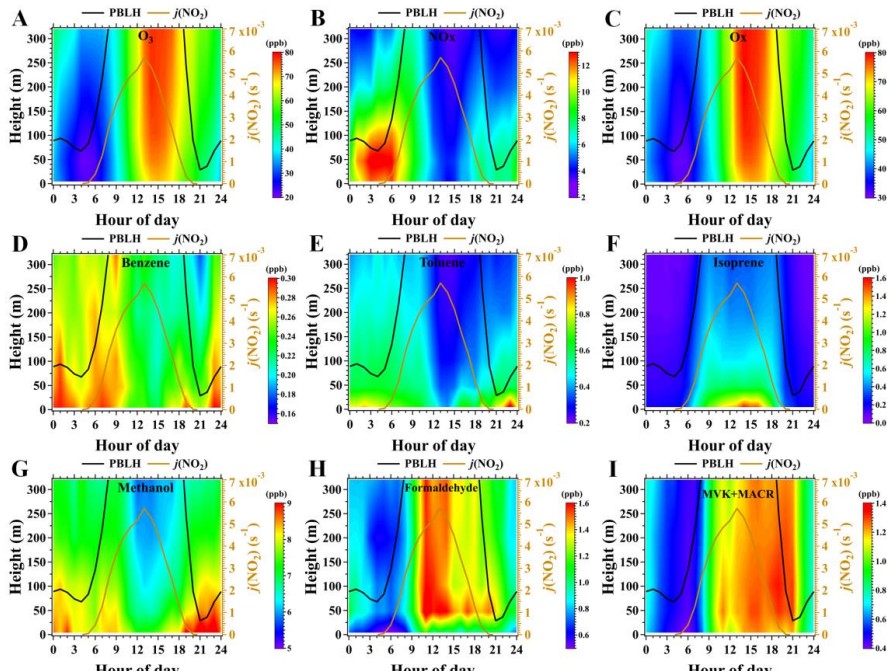

**Figure 2.** Average diurnal and vertical variations in mixing ratios ozone, NOx, Ox
($O_3$+NO_2$), and six selected VOC species along with the average diurnal profiles of
PBLH and $j$(NO_2$) during the campaign. The figures were obtained by linearly
interpolating the data at the five inlet heights on both altitude and temporal scales.

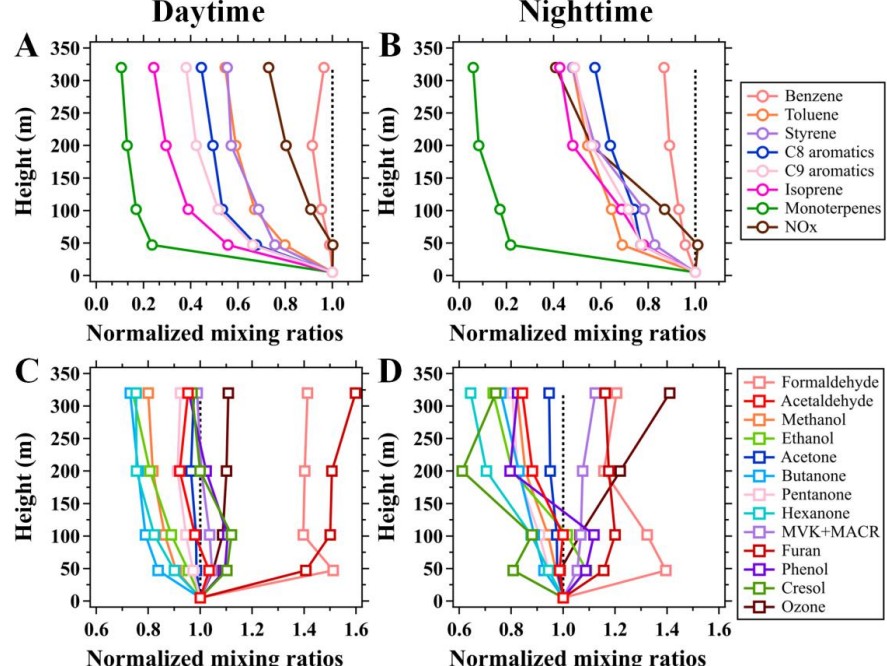

**Figure 3.** Average vertical profiles of (A-B) NMHCs and NOx, (C-D) OVOCs and $O_3$

during the daytime (11:00-16:00 LT) and nighttime (23:00-04:00 LT) of the campaign.

The mixing ratios of the chemical species measured above 5 m are normalized to those

at 5 m.

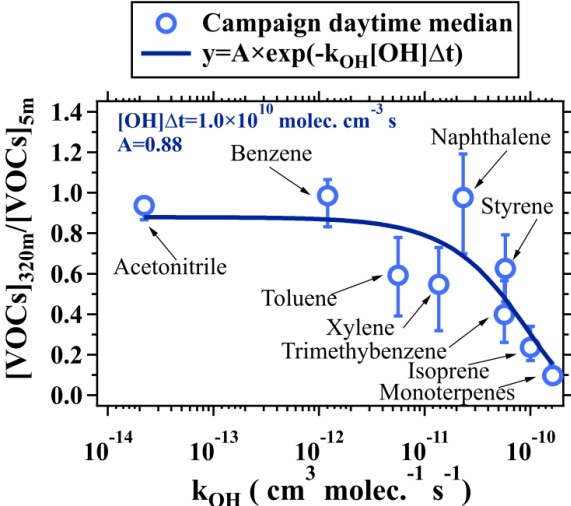

**Figure 4.** The change in ratios of NMHC concentrations (including acetonitrile) between 320 m and 5 m as a function of $k_{OH}$. The vertically-resolved measurements of VOCs made on the BMT in daytime during the campaign were used for analysis. Hollow markers represent median values and error bars indicate the range between 25th and 75th percentiles.



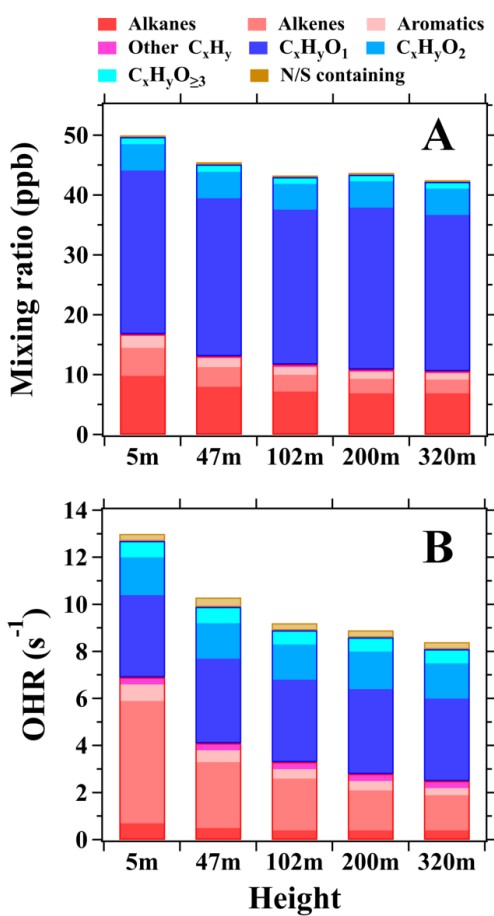

**Figure 5.** (A) Mean mixing ratios and (B) OHRs of different VOC categories at the five

inlet heights in daytime during the campaign.

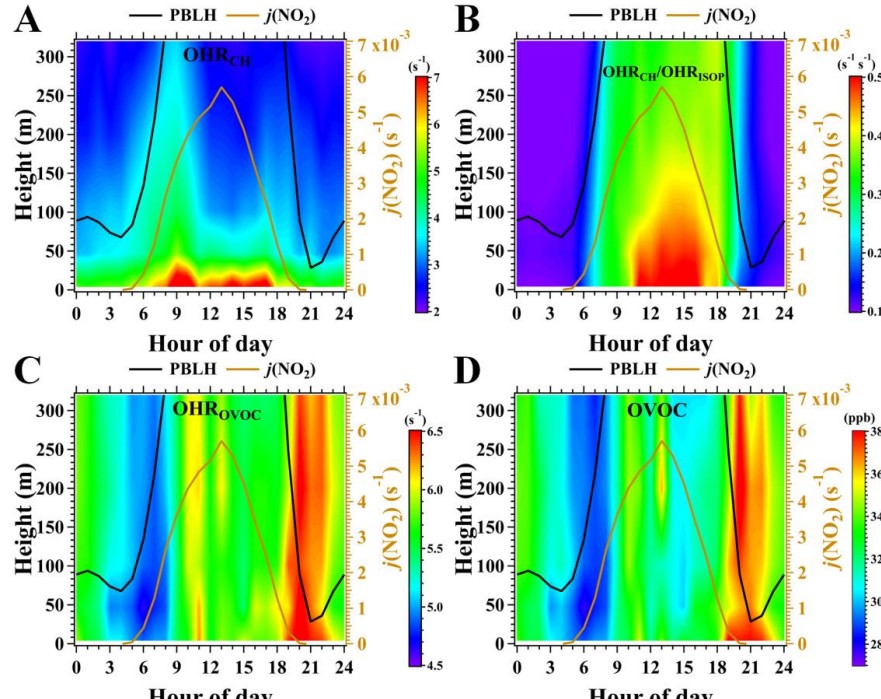

**Figure 6.** (A-B) Average diurnal and vertical variations in OHRs of $C_xH_y$ and the OHR ratios of isoprene to $C_xH_y$ (OHR$_{ISOP}$/OHR$_{CH}$) during the campaign. (C-D) Average diurnal and vertical variations in mixing ratios and OHRs of OVOC. ISOP refers to isoprene. The figures were obtained by linearly interpolating the data at the five measurement heights on both altitude and temporal scales.



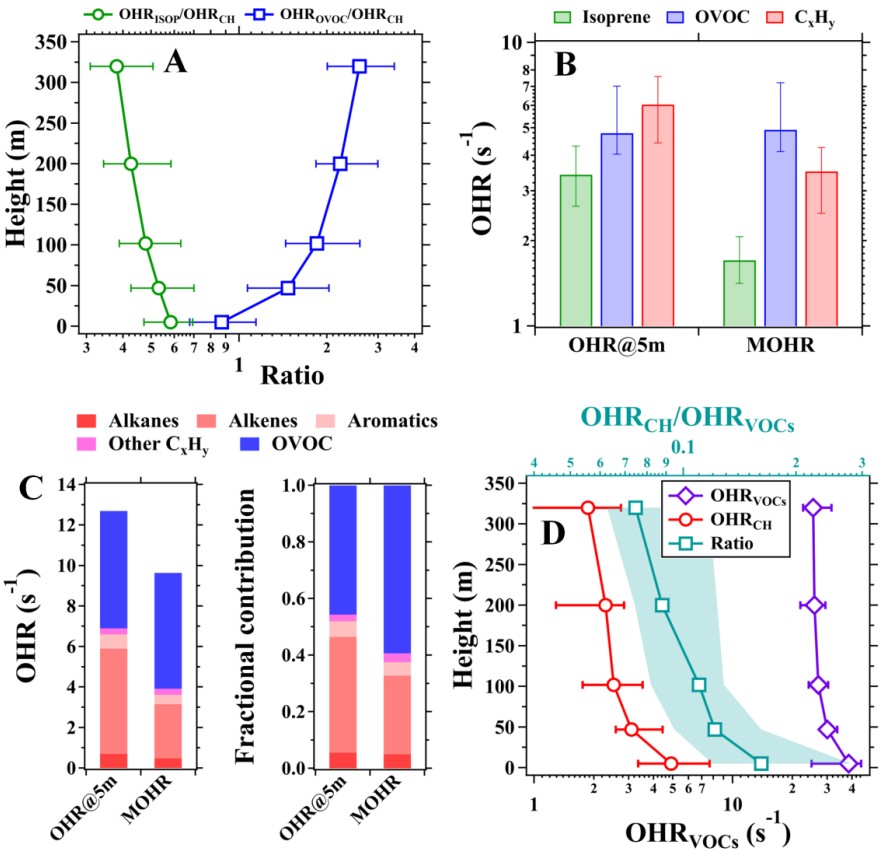

**Figure 7.** (A) Average vertical profiles of OHR ratios of isoprene to $C_xH_y$ ($OHR_{ISOP}/OHR_{CH}$) and OVOC to NMHC ($OHR_{OVOC}/OHR_{CH}$). (B) Median values of the OHR at 5 m and the mean OHR (MOHR) between 5 m and 320 m for isoprene, OVOC, and $C_xH_y$. (C) Mean contributions of different VOC categories to the MOHR below 320 m and the OHR at 5 m. (D) Vertical profiles of the measured $OHR_{VOCs}$ and the calculated $OHR_{CH}$ (bottom axis) and the $OHR_{CH}/OHR_{VOCs}$ ratios (top axis) during July 28-31, 2021. The data used for analysis in panels A-D was within the time window from 11:00 to 16:00 LT during the campaign. Markers in panels A and D represent median values. Shaded areas and error bars in panels A, B, and D indicate the range between 25th and 75th percentiles.

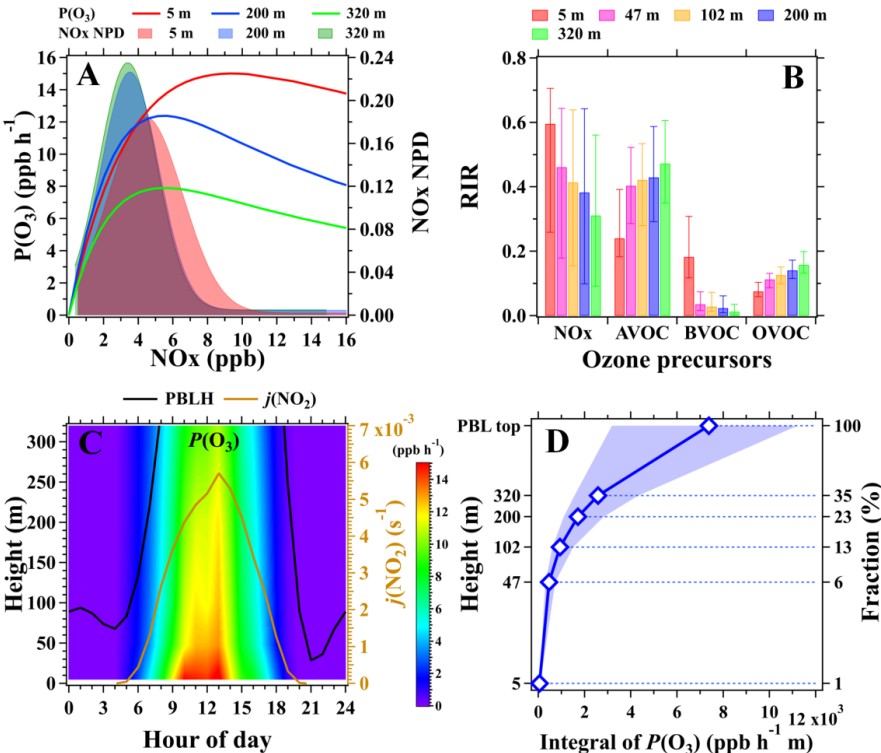

**Figure 8.** (A) Left axis: average dependence of P(O$_3$) on NOx concentrations in daytime during the campaign; Right axis: normalized probability density (NPD) of NOx mixing ratios in daytime at the three inlet heights. (B) Median RIR values of photochemical ozone formation to changes in NOx, AVOC (NMHCs excluding BVOC), BVOC (isoprene), and OVOC (nine OVOC species in Table S1) at the five inlet heights; Error bars indicate the range between 25$^{th}$ and 75$^{th}$ percentiles. (C) Average diurnal and vertical variations in $P$(O$_3$) during the campaign; The figure was obtained by linearly interpolating the data at the five measurement heights on both altitude and temporal scales. (D) The vertical profile of the integral of $P$(O$_3$) in daytime during the campaign; Markers indicate median values and Shaded areas indicate the range between 25$^{th}$ and 75$^{th}$ percentiles.