# Peer review of "Vertical changes in volatile organic compounds (VOCs) and"

_EGUsphere, 2024_

## Author Comment (AC1)

* * *
**Response to Reviewer #1**
* * *
**1.** Li et al. present vertically resolved VOC and oVOC measurements from Beijing, China. The measurements were taken in a 325 m tall tower switching between several heights from near the surface (5 m) to the top of the tower (320 m). The authors then use these measurements to model and calculate a variety of atmospheric parameters including OH reactivity, photochemical ozone production, and vertical gradients in VOC concentrations. While the analysis performed is not necessarily novel, the authors present fresh measurements and thorough analysis which contributes to our understanding of vertical VOC distributions and the role of boundary layer dynamics and vertical mixing in ozone formation which is desirable. One of the most important conclusions of the article is the importance of oVOCs at higher altitudes and their ability to contribute to ozone formation which can then affect surface concentrations. Only 35% of the PBL ozone is produced in the first 320 m. The measurements and analysis are sound and support the conclusions of the paper. The article is well written and properly referenced. I believe the article can be published as is. However, isoprene plays a significant role in the chemistry observed and the results and yet the mechanism used to model isoprene, the MCMv3.3.1, is significantly outdated. Wennberg et al. (https://doi.org/10.1021/acs.chemrev.7b00439) published an updated mechanism for isoprene and its oxidation products. The mechanism can be retrieved at (https://data.caltech.edu/records/x88rk-wca37) and it is readily integrated with F0AM. The updated mechanism includes isomerization reactions, additional oVOCs, better representation of isoprene derived organic nitrates, updated reaction rates among many changes, which could significantly affect some of the quantitative results of the modeling work. I do not believe the overall conclusions of the article would change, which is why I support publication as is, however, using an updated mechanism would improve the analysis significantly.

**Reply:** We appreciate your valuable comments and suggestions, which are very

helpful for the improvement of our manuscript.

We have carefully read the paper "Gas-Phase Reactions of Isoprene and Its Major Oxidation Products" (Wennberg et al., 2018) recommended by the reviewer. The chemical mechanism of isoprene in this paper primarily includes the reactions of isoprene and its products with chlorine radicals (e.g., $C_5H_8$+Cl, MCK+Cl, MACR+Cl) and the reactions of hydroxymethylperoxide (HMHP) with OH radicals. The MCMv3.3.1 model has integrated most of the mechanisms reported in the literature. The isoprene-related mechanisms mainly based on "The MCM v3.3.1 degradation scheme for isoprene" (Jenkin et al., 2015) and the related parameters are recommended by IUPAC (https://iupac.aeris-data.fr/en/home-english/).

According to the results of previous studies, the oxidation initiated by chlorine radicals is an only minor sink for isoprene, and this oxidation pathway is likely important in the marine boundary layer. However, our research is conducted in inland areas and we thus speculate that the oxidation of isoprene by chlorine radicals is minor. When considering reactions with oxidants such as OH, $O_3$, and $NO_3$, there might be some differences in reaction rates or branching ratios between the two mechanisms. However, most of the reaction rates used in the two models are still predominantly based on those recommended by IUPAC. The purpose of this paper is to compare the vertical distributions of different VOC species and their impacts on ozone formation. The addition of the above reactions may not affect the existing results. We are still very grateful for your advice and our future studies regarding the simulation of isoprene and other VOCs will employ this latest mechanism.

Small typos:

**2.** Line 79: "the ozone formation regime *like* undergoes…"

**Reply:** Thank you for pointing out this mistake. It should be "likely" here and we have corrected it in the revised manuscript. [see P: 4; L: 81-83]

"*The ozone formation regime likely undergoes significant transitions from the ground to the upper boundary layer (Li et al., 2024; Liu et al., 2024a)*"

**3.** Line 433: "approximately 9.5 ppb at 5 m to 5.0 *m* ppb at 320 m…"

**Reply:** Thank you for pointing out this mistake. It should be "5.0 ppb" here and we have corrected it in the revised manuscript. [see P: 17; L: 434-436]

*"The critical NOx mixing ratios decreased from approximately 9.5 ppb at 5 m to 5.0 ppb at 320 m, primarily caused by the decreases in both NOx concentrations and the OHRs of VOCs."*

---

## Author Comment (AC2)

**Response to Reviewer #2**

**1.** This paper presented a very comprehensive vertical measurement of air pollutants and analysis of the gradient of ozone and precursors, as well as the response of vertical ozone formation sensitivity on the vertical gradients of VOC and OVOCs, this work provides a good example and new insight to understand the ozone pollution, I would like to recommend this paper publish in ACP subjects some minor suggestions.

**Reply:** We appreciate your valuable comments and suggestions, which are very helpful for the improvement of our manuscript.

**2.** Lines 27-35, the abstract is very confusing and hard to follow, I suggest the authors try to rephrase these sentences.

**Reply:** We appreciate your valuable suggestions and we have rephrased the abstract to make it clearer in the revised manuscript. [see P: 2; L: 21-40]

*"Volatile organic compounds (VOCs) play crucial roles in regulating the formation of tropospheric ozone. However, limited knowledge on the interactions between vertical VOC variations and photochemical ozone formation in the planetary boundary layer (PBL) has hindered effective ozone control strategies, especially in large cities. In this study, we investigated the vertical changes in concentrations, compositions, and key driving factors of a large suite of VOCs using online gradient measurements taken from a 325 m tall tower in urban Beijing. The impact of these vertical VOC variations on photochemical ozone formation were also analyzed using box model simulations. Our results indicate that VOCs exhibited distinct vertical variation patterns due to their differences in sources and chemical reactivities, along with the diurnal evolution of the PBL. During daytime, reactive VOCs (e.g., hydrocarbons) are rapidly oxidized as they mix upwards, accompanied by the formation and accumulation of oxygenated VOCs (OVOCs) in the middle and upper layers. In addition, the photochemical formation of ozone responds positively to changes in both*

*NOx and VOCs. As a result, the production rate of ozone declines with height due to the simultaneous decreases in concentrations of reactive VOCs and NOx, but remains high in the middle and upper layers. The strong production of ozone aloft is primarily driven by the presence of high OVOCs concentrations. Therefore, careful consideration should be given to the vertical variations in both photochemical ozone production rates and formation regimes in the whole PBL when developing regional ozone control strategies*"

**3.** "The strong production of ozone aloft is primarily driven by high concentrations of OVOCs and hydroxyl radicals", the OH is modeled rather than observed, thus I suggest removing OH here to make it more conservative.

**Reply:** We appreciate your valuable suggestions. In our study, OH concentrations at different altitudes were obtained from box model simulations. In order to make the analysis more rigorous, OH radicals have been removed here [see P: 2; L: 37-38].

*"The strong production of ozone aloft is primarily driven by high concentrations of OVOCs."*

**4.** The vertical profile of many VOC species concentrations is estimated and scaled by another site measurement as described by the SI file, I think this part would bring large uncertainty to the box model as well as the following results, I suggest the author add more discussions about the uncertainties.

**Reply:** We appreciate your valuable comments and suggestions. In our study, uncertainties of the method mainly come from the following three aspects. First, the differences in source emissions of VOCs, which will significantly change the concentration ratios of VOC species between the two sites. In urban Beijing, vehicular exhausts are the predominant source of ambient VOCs, which could be supported by the average concentration ratio of toluene to benzene (T/B=1.45, a T/B ratio of around 2 usually indicates emissions of vehicular exhausts), as shown in Fig. S2. Therefore, there are no significant differences in sources of VOCs between the two sites.

[Figure]

**Figure S2.** Scatter plot of toluene to benzene mixing ratios at ground level at the BMT site during the field campaign.

Second, the differences in vertical concentration gradients of VOC species. In daytime boundary layer, NMHCs are mainly removed by reactions with OH radicals. The vertical concentration gradients of NMHCs have strong dependence on their reaction rate constants with OH radicals, as shown in Fig. 4. In nighttime, NMHCs are mainly removed by reactions with $O_3$ and $NO_3$ radicals, which are much smaller than those with OH radicals and thus have relatively minor impacts on the vertical concentration gradients of NMHCs. In this study, the concentrations of unmeasured NMHCs at different altitudes were estimated using the concentrations of measured aromatic species with similar $k_{OH}$ values. By using this method, the effects of the differences in VOCs vertical gradients on the estimation of unmeasured NMHCs can be minimized.

Third, the differences in effects from advection transport. This is also the most difficult point to manage for estimating NMHC concentrations. Considering the short spatial distance (5 km) between the two sites, effects of transport on the change in concentrations of NMHCs at the two sites should have minor differences. Therefore, this approach will bring minor uncertainties in estimating concentrations of NMHCs at different altitudes on the BMT site. We have provided related discussions in the revised SI file [see P: 2-4 in SI].

*"The uncertainties associated with the method primarily stem from three key aspects. Firstly, variations in source emissions of VOCs can significantly alter the concentration ratios of different VOC species between the two sites. In urban Beijing, vehicular exhausts are the dominant contributor to ambient VOCs, a fact corroborated by an average toluene-to-benzene ratio (T/B) of 1.45—a T/B ratio around 2 typically signifies vehicle emissions, as illustrated in Figure S2. Consequently, it can be inferred that there are no substantial disparities in VOC sources between the two sites.*

*Secondly, differences in vertical concentration gradients among VOC species pose another uncertainty. During daylight hours within the boundary layer, NMHCs are primarily removed through reactions with OH radicals. The magnitude of these vertical gradients is strongly influenced by the reaction rate constants of NMHCs with OH radicals, as depicted in Figure 4. Conversely, at night, NMHC removal mainly occurs via reactions with $O_3$ and $NO_3$ radicals, which have much lower reaction rates compared to OH and hence exert lesser impact on the vertical distribution of NMHCs. To mitigate this issue, our study employs a strategy where the concentrations of unmeasured NMHCs across various altitudes are inferred from those of measured aromatic compounds with analogous $k_{OH}$ values. By using this method, the effects of the differences in VOCs vertical gradients on the estimation of unmeasured NMHCs can be minimized.*

*Lastly, advection transport effects introduce variability, representing the most challenging uncertainty to address in estimating NMHC concentrations. Given the relatively short geographical separation (5 km) between the two sites, however, it is reasonable to assume that transport-induced changes in NMHC concentrations would exhibit minimal differences. Thus, while acknowledging this potential source of uncertainty, we contend that its impact on estimating NMHC concentrations at different altitudes on the BMT site would be marginal."*

**5.** Line 213-215, why the dry deposition rate be set to the value of 0.27 cm $S^{-1}$ at the surface, and the other heights set to zero?

**Reply:** We appreciate your valuable comments and suggestions. Dry deposition of chemical species in the atmosphere is primarily influenced by the aerodynamic resistance, the quasi-laminar sublayer resistance through the leaves, and the overall canopy resistance at the surface of the vegetation canopy (Nguyen et al., 2015; Zhang et al., 2003). Therefore, we assume that the deposition of chemical species mainly occurs at or near the ground. The model simulations at higher altitudes are also conducted using a box model, where the influence of canopy resistance is negligible. Therefore, the deposition effects of chemical species are not considered in the model simulations at higher altitudes.

*References:*

*Nguyen, T. B., Crounse, J. D., Teng, A. P., St. Clair, J. M., Paulot, F., Wolfe, G. M., and Wennberg, P. O.: Rapid deposition of oxidized biogenic compounds to a temperate forest, Proceedings of the National Academy of Sciences, 112, E392, 10.1073/pnas.1418702112, 2015.*

*Zhang, L., Brook, J. R., and Vet, R.: A revised parameterization for gaseous dry deposition in air-quality models, Atmos. Chem. Phys., 3, 2067-2082, https://10.5194/acp-3-2067-2003, 2003.*

**6.** Line 338-342, how are these species calibrated?

**Reply:** We appreciate your valuable comments. In our study, sensitivities of 39 VOC species measured by the PTR-ToF-MS were calibrated using a gas standard. Sensitivities of the remaining species measured by the PTR-ToF-MS were estimated based on their relationships with reaction kinetics of $H_3O^+$ (Figure R1), which has been detailed in our previous paper (Wu et al., 2020). This information is provided in section 2.1 in the manuscript. [see P: 7; L: 158-161]

*"A gas standard containing 39 VOC species was used to calibrate the PTR-ToF-MS daily. Sensitivities for the remaining species were determined based on reaction kinetics of the PTR-ToF-MS (Wu et al., 2020)."*

[Figure]

Figure R1 Corrected sensitivities as a function of kinetic rate constants for proton transfer reactions of $H_3O^+$ with VOCs. The dashed line indicates the fitted line for blue points. The red points not used as these compounds (formaldehyde, methanol, and ethanol) are known to have lower sensitivities (Wu et al., 2020).

***References:***

*Wu, C., Wang, C., Wang, S., Wang, W., Yuan, B., Qi, J., Wang, B., Wang, H., Wang, C., Song, W., Wang, X., Hu, W., Lou, S., Ye, C., Peng, Y., Wang, Z., Huangfu, Y., Xie, Y., Zhu, M., Zheng, J., Wang, X., Jiang, B., Zhang, Z., and Shao, M.: Measurement report: Important contributions of oxygenated compounds to emissions and chemistry of volatile organic compounds in urban air, Atmos. Chem. Phys., 20, 14769-14785, https://doi.org/10.5194/acp-20-14769-2020 2020.*

**7.** How about the set of HONO in the model, this is critical to the ozone sensitivity and OH field, especially since the HONO may be higher at the ground than aloft, which would decrease the vertical gradient of OH.

**Reply:** We appreciate your valuable comments and suggestions. We agree with your opinion that HONO plays a vital role in contributing to the budgets of OH radicals and thus affecting the ozone formation. In our study, HONO concentrations were not measured during the field campaign and thus are not constrained in the box model. In the setup of the model, the source of HONO is the product of OH+NO and the sink is

its photolysis. HONO has complicated sources, such as direct emission of combustion sources, heterogenous formation, and gaseous formation. Previous studies have reported that HONO concentrations will be underestimated without adding additional sources in the model. In this condition, 2% of measured $NO_2$ concentrations was generally used to represent HONO concentrations. The vertical observation results of some previous studies also indicate that HONO concentrations have a decreasing trend with height, but the ratio of HONO to $NO_2$ remains at ~2% (Wong et al., 2012; Tan et al., 2021). Therefore, we have conducted sensitivity tests by constraining HONO at a ratio of 2%×$NO_2$ at three altitudes (namely 5, 200, and 320 m) to examine the effects of HONO setup in the model on the results regarding photochemical ozone formation. As shown in Fig. R2, the turning point of the $P(O_3)$ curves changes insignificantly between the modeling results with and without the constraint of HONO, implying that the constraint of HONO or not in the model has minor impacts on the identification of ozone formation regimes. With the constraint of HONO, the increase in modeled $P(O_3)$ is significantly enhanced in scenarios with high NOx concentrations. These situations occur rarely in urban Beijing in summertime (Fig. 8A). Therefore, we think the constraint of HONO may have a minor impact on the results of this study.

***References:***

*Wong, K. W., Tsai, C., Lefer, B., Haman, C., Grossberg, N., Brune, W. H., Ren, X., Luke, W., and Stutz, J.: Daytime HONO vertical gradients during SHARP 2009 in Houston, TX, Atmos. Chem. Phys., 12, 635-652, 10.5194/acp-12-635-2012, 2012.*

*Tan, Z., Wang, H., Lu, K., Dong, H., Liu, Y., Zeng, L., Hu, M., and Zhang, Y.: An Observational Based Modeling of the Surface Layer Particulate Nitrate in the North China Plain during Summertime, Journal of Geophysical Research: Atmospheres, 126, e2021JD035623, https://doi.org/10.1029/2021JD035623, 2021.*

[Figure]

Figure R2 Average dependence of P(O₃) on NOx concentrations in daytime during the
campaign. The results are obtained from box model simulations.

---

## Author Response (AR2)

* * *
**Response to the editor**
* * *
**1.** The manuscript is of excellent technical quality, as pointed out by all reviewers. However, I would like to ensure the manuscript fits with the ACP author guidelines. Specifically the conclusions section is somewhat incomplete. I would like to ask the authors to provide a more detailed discussion of comparisons, context, limitations and broader implications. It would make the manuscript even stronger.

**Reply:** We appreciate your valuable comments and suggestions, which are very helpful for the improvement of our manuscript. Based on your suggestions, we have rephrased the conclusions section and provided more discussions on the context of our study, comparisons with previous works, implications for future studies on secondary air pollution, and the limitations of our study. [see P 19-20, L 484-524]

"*The inadequate vertical distribution data of volatile organic compounds (VOCs) poses a significant barrier to fully comprehending the mechanisms underlying photochemical ozone formation and devising effective mitigation strategies. To address this concern, we made vertical gradient measurements of VOCs, NOx, and ozone based on a 325 m tall tower in urban Beijing during the summer of 2021. This study offered more exhaustive and nuanced insights into the vertical variability of VOCs compared to previous studies. Our findings underscored that the vertical variations of VOCs were strictly regulated by the diurnal evolution of the PBL and chemical processes. In daytime, reactive NMHCs were rapidly oxidized when they were mixed upward along with the formation of OVOCs. As a result, concentrations of NMHCs decreased with height and many of OVOC species increased with height. OVOC species played more significant roles in regulating the photochemical ozone formation in urban regions aloft.*

*Model simulations unveiled that the photochemical formation of ozone belongs to the transition regime in the lower PBL and became more sensitive to changes in the concentrations of AVOCs and OVOCs with height. With the further increase in height,*

*the photochemical formation of ozone may change to the NOx control regime due to the total OHR of VOCs decreased much slower than NOx concentrations. $P(O_3)$ exhibited decreasing tendencies with height due to coupled declines in concentrations of NOx and VOCs. $P(O_3)$ still remained large in high altitudes, likely driven by high OVOC concentrations. This implies that the bulk of ozone formation occurs within the middle and upper strata of the PBL rather than proximate to the ground surface. Therefore, regional ozone control strategies necessitate meticulous consideration of vertical gradients in $P(O_3)$ and the varying regimes of photochemical ozone formation throughout the entire PBL.*

*The vertical variations in concentrations and compositions of VOCs significantly influence ozone formation. In addition, the vertical changes in chemical reaction environments (e.g., temperature, humidity, and solar radiation) and concentrations of other chemicals (e.g., particulate matters, NOx, ozone) can also impact the degradation pathways of VOCs. These factors also affect the formation pathways and production yields of other secondary air pollutants, such as formic acid, isocyanic acid, and secondary organic aerosol. This is particularly crucial for the highly reactive NMHCs in urban areas with complex anthropogenic and biogenic emissions.*

*Limitations of our study include the confinement of measurements below 320 meters due to the tower's height, leaving the mid and upper daytime PBL's VOC distributions and chemistries unexplored. Additionally, the absence of measurements for some key chemical species like HONO, organic aerosol components, and reactive halogen species might have implications for the accuracy of our box model results. Future endeavors could integrate multiple observational techniques to capture a broader vertical scope and a more comprehensive suite of species, thereby enhancing our understanding of how vertical variations in VOC chemistry impact secondary pollution formation.*"